# Effectiveness and Health Outcomes of Collaborative Nurse Prescribing for Patients Anticoagulated with Antivitamin K in Primary Care: A Study Protocol

**DOI:** 10.3390/healthcare12030347

**Published:** 2024-01-30

**Authors:** Juan Carlos Palomo Lara, Joan Carles March-Cerdà, José Antonio Ponce-Blandón, Manuel Pabón-Carrasco, Nerea Jiménez-Picón

**Affiliations:** 1Seville Primary Care District, Red Cross and San Juan de Dios Nursing School, Faculty of Nursing, Physiotherapy and Podiatry, University of Seville, Av. Jerez, 41013 Seville, Spain; juanc.palomo.sspa@juntadeandalucia.es; 2Andalusian School of Public Health, 18011 Granada, Spain; 3General Manager Sports Medicine Center Junta de Andalucía, Faculty of Nursing, Physiotherapy and Podiatry, University of Seville, 41004 Sevilla, Spain; 4Faculty of Nursing, Physiotherapy and Podiatry, University of Seville, 41004 Sevilla, Spain

**Keywords:** nursing, collaborative prescription, evaluation of the efficacy and effectiveness of interventions, outcome and process assessment (health care), evaluation of results of therapeutic interventions, medication therapy management, anticoagulants

## Abstract

The development of collaborative nurse prescribing (NP) in Andalusia (Spain) in 2018 gives us the opportunity to measure the impact of this practice. Scientific evidence indicates that prescribing is not more costly when performed by nurses and, in fact, is more economical in some cases. The aim of this study is to determine the effects of NP on the effectiveness, health outcomes and adverse events related to prescribing including in the follow-up of patients treated with antivitamin K oral anticoagulants in primary care (PC) by nurses. The design is a randomized clinical trial. The population comprises 1208 anticoagulated patients in 2019. The sample size calculation considers an alpha error of 0.05, a power of 99% and an effect size of 0.5, resulting in 127 users per group. Therefore, a total sample of 254 participants is needed. However, as the project intends to treat patients it will include the universal sample that meets the criteria in the two health centers participating in the study, with 575 participants in total. Data collection was carried out in the PC District of the Alamillo-San Jerónimo Clinical Management Unit of Sevilla for one year from January 2020. Data analysis is performed using the SPSS Statistics 25 package. We intend to study if nurse collaborative prescription in the follow-up and management of patients taking antivitamin K oral anticoagulants in PC is as effective as the traditional approach to follow-ups carried out by a family physician.

## 1. Introduction

The World Health Organization (WHO) has been warning countries about the increase in chronic diseases and existing comorbidities as well as the resulting increased spending on services and benefits, providing them with recommendations to implement strategies to address chronic diseases. These include strengthening the role of primary care (PC) as well as the management and follow-up of chronic patients where nurses are key [1].

Effectively sharing the care of stable chronic patients between physicians and nurses delivers effective results in user satisfaction, the rationalization of costs and care time. In this approach, nurses play a pivotal role in the follow-up of patients and their families by responding to their health needs and becoming involved in their treatment (collaborative prescribing with the physician) and results (complementary tests properly rationalized and protocolized, indication and results) [2,3].

In all healthcare scenarios, especially in those that consume the most resources, improving efficiency may involve restructuring healthcare to provide a more versatile system that is adapted to each level of complexity and connects the patient to the most appropriate healthcare provider in each case. This entails developing the nursing competency framework further so that the responsiveness of healthcare professionals is not hindered by care models that no longer support the new needs of patients related to chronicity, clinical complexity, vulnerability or fragility [4].

In the management, monitoring and follow-up of anticoagulated patients, the current trend is one that is moving toward a mixed model in which the follow-up of the most complex anticoagulated patients is carried out by the hematology services of hospitals, while PC professionals monitor the treatment of stable anticoagulated patients (approximately 70%) [5].

An analysis of the outcomes of the substitution of physicians for nurses in certain clinical interventions can be found in the literature consulted. Horrocks et al. conducted a systematic review in 2002 in which they concluded that increasing the availability of PC nurses was likely to improve patient satisfaction and the quality of care received. Among the limitations, studies with a longer follow-up time are recommended [6]. 

Laurant et al. conducted a systematic review in 2008 to investigate the impact of nurses as substitutes to PC physicians on patient outcomes, healthcare processes, resource utilization, direct (service) costs and indirect (societal) costs. The authors concluded that many countries attempt to shift the provision of PC from physicians to nurses to reduce the demand for physicians and improve the efficiency of healthcare. Nurses working as substitutes are expected to be able to provide healthcare of a similar quality to that of physicians at a lower cost. This review found that properly trained nurses can provide patients with the same high-quality care as that of PC physicians and achieve similar health outcomes. However, the available research was quite limited [7].

Internationally, in the USA (EE.UU.) and Finland, anticoagulated patients have been followed up by nurses for some time using models similar to those being developed in Spain, such as that using joint follow-up protocols, with good outcomes in patient management [8,9]. 

Studies comparing traditional follow-up by PC physicians with follow-up by nurses, either in health centers or in anticoagulation clinics, show health outcomes with more time in therapeutic range, fewer blood tests [8] and reduced costs [9]. 

A wide range of care models can be applied to patients undergoing oral anticoagulation therapy with VKAs [10]. Some models have the PC physician independently manage oral anticoagulation therapy. Patients in this scenario are given the responsibility to undertake laboratory tests, which are prescribed by the physician, such as prothrombin time and INR ones. Alternatively, patients are able to perform their own self-testing rather than relying on an external laboratory. In either case, the physician must be informed of the results, will decide on dosage adjustments or further treatment and conduct follow-up appointments over the phone [10].

Anticoagulation clinics, which are managed by either pharmacists or nurses, are yet another approach to care. Within this model, practitioners strive to provide the organized and thorough management of oral anticoagulation therapy. Evidence suggests that providing care through anticoagulation clinics achieves a number of objectives, which are according to Francavilla [11] as follows:-Improve the assessment and monitoring of patients being treated with oral anticoagulation with VKAs.-Improve dosage adjustment with greater accuracy.-Encourage patient adherence to the prescribed treatment regimen.-Empower patients to take greater responsibility for self-care through comprehensive health education.-Reduce complications, hospitalizations and urgent care visits related to anticoagulation therapy.-Increase patient satisfaction and improve quality of life.-Decrease physicians’ workloads.-Establish a framework to assess the outcomes and quality of outpatient anticoagulation management services by nurses.

There are, however, some measurement constraints of specific outcome data or further evidence [10].

Guidelines were created for nurses to monitor therapy through laboratory testing in collaboration with a physician, which is similar to how it is managed in the field. This is undertaken using guidelines supported by evidence and consensus from the American Colleges of Chest Physicians. These were written to encourage autonomous decision making by certified nurses based on individual patients’ needs, which is similar to the collaborative prescribing manual for patients on oral anticoagulation [12].

Francavilla explains the tasks that are carried out by certified nurses in the management of health education, which are undertaken in the field both in the process of INR dosing by the nurse or in the monitoring of chronic patients in the nurse’s office by their family nurse [11].

The suggested monitoring is similar to the guidelines in the oral anticoagulation manual [12,13] that was used as a baseline in our study protocol. Patients are seen at least once a month if they are fairly stable. However, as indicated through the protocols, the patient has to be seen at a higher frequency when dosage changes occur. This entire process is recorded in the medical record. Moreover, a doctor is available for telephone consultation if necessary [11,12].

The comparison of different models of care for anticoagulation treatment management is often complex [14]. This is due to the lack of enough data to compare and contrast. A New York clinic conducted a study on a sample of 20 patients with a minimum of 10 clinic visits in 2006 compared with the same 20 patients in 2005, with the former undergoing a regular follow-up. Proper monitoring was reflected in the percentage of their INRs within the therapeutic range of +0.2. In 2005, prior to the study, the therapeutic range of the 20 patients was 59%, while these same 20 patients achieved a therapeutic range compliance rate of 73% in 2006. Therefore, a 14% increase in the patients in the range was achieved with the newly developed management model. A target of 70% compliance with the INR range of +/−0.2 was set. A compliance rate of 68% to 71% was achieved [14].

Patient care management in anticoagulation clinics is in need of further evaluation as little research is available to demonstrate that this care model improves patient outcomes and is also cost-effective. The majority of the existing literature is in the form of narrative testimonials and analyses of single data repositories. Additional randomized controlled trials would actually provide a better evidence-based literature to support the perceived improvement of patient outcomes in anticoagulation clinics [11].

In 2008, researchers aimed to compare the efficacy of an algorithm developed to be used in a nurse-managed outpatient anticoagulation clinic with the use of clinical judgement without formalized guidelines. This was undertaken using a retrospective study that compared 179 patient visits in September 2007 with 206 in 2009 at the Johns Hopkins Outpatient Anticoagulation Clinic. An algorithm that incorporated “removable causes” was created to help nurses in their decision making, for instance, when adding and removing new medication that interacts with Warfarin [15].

There were higher percentages of an absence of changes versus changes in warfarin dosages, which reflect stable dosing patterns in both years. Chi-square analysis showed no statistical significance in the relationship between dosage changes and the year. Nevertheless, a significant correlation was found between removable causes and the year. This indicates an improvement in the recording of removable causes in 2009 [11].

As a limitation, the authors indicate the need for a larger randomized prospective population sample to more thoroughly evaluate the effect of the algorithm on time in therapeutic range (TTR) [11].

In 2015, a prospective study was conducted with patients who had just started VKA treatment for AF being followed up by nurses in our country. A comparison group was established that consisted of AF patients who were already on VKA treatment for 6 months according to standard clinical practice. Moreover, the anticoagulation quality assessment was performed after 6 months, with efficacy and safety outcomes being recorded during follow-ups [16].

The sample included a total of 223 patients (nurse follow-ups: 107; standard care: 116). The average time in therapeutic range and the proportion of INR within therapeutic range were found to be similar in both groups. During a follow-up of 2.06 years (ICR 1.01–2.94), 64 patients (28.7%) switched to new oral anticoagulants. A higher proportion of patients switched in the nurse-led clinic (37.4%) compared with the usual care group (20.7%) (*p* = 0.006). Furthermore, these patients needed less time to change (2.0 [ICER 0.7–2.9] vs. 6.0 [ICER 3.7–11.2] years, *p* < 0.001). Notably, the annual rate of transient ischemic attack (TIA) was significantly lower in the nurse follow-up group (0.47%/year vs. 3.88%/year, *p* = 0.016) with no difference in safety outcomes [16]. Authors conclude an AF patient follow-up by nurses can provide comprehensive patient centered care and follow-ups. Furthermore, this may be associated with a decrease in the incidence of TIA and with no increase in hemorrhagic complications. The authors recommend further studies to validate these results [16].

Nurses with varying levels of undergraduate, graduate and pathology-related job-specific education were able to produce prescribing outcomes comparable to those of physicians. Non-physician prescribers often rely on physician support to facilitate a collaborative practice model [17].

Anticoagulants have been used for a long time to prevent thrombosis in certain scenarios, such as for patients with atrial fibrillation (AF), valve diseases and valve replacements, thromboembolic processes, acute myocardial infarction, peripheral artery disease and coronary stents [18]. Among these, AF is the most common chronic sustained arrhythmia, with its prevalence doubling with each decade of life from 0.55% at age 50–59 to 9% at age 80–89 [19]. It is the most frequent pathology requiring treatment with antivitamin K (AVK) anticoagulants. 

In Spain, more than one million patients use them for this indication. The aim of oral anticoagulant therapy (OAT) is to lengthen the clotting time to an effective and safe interval (therapeutic range) in which thrombus development is prevented without causing a risk of hemorrhage. In non-anticoagulated individuals, the international normalized ratio (INR) is close to or equal to 1. The ideal INR for each anticoagulated patient may vary, usually ranging between 2 and 3, or being slightly higher, depending on the characteristics of each individual and the cause of treatment [18].

The anticoagulant effect is insufficient if the INR is below the therapeutic range, and, conversely, if it is much higher there is an increased risk of bleeding. Nurses are responsible for the care of these patients in both PC and hospital care settings, with the ultimate goal of empowering the patient and managing their process and disease.

In complex treatments, as is the case here, assessing compliance, coping skills and understanding the warning signs and symptoms of complications are of vital importance to ensure patient safety and the expected health outcomes [18]. In recent years, standardization in the measurement of prothrombin time (INR), the use of lower therapeutic ranges for various indications (lower frequency of hemorrhagic complications), the increase in the indications for OAT and the appearance of portable coagulation analyzers have meant that more and more patients are beginning to be monitored by PC teams through this treatment with greater accessibility and more comprehensive patient care [5].

With regard to the outcomes in terms of efficiency and the improved sustainability of the system, the nurse’s role as a prescriber has been introduced by the Andalusian Health Service (SAS) as a strategy to control the increase in pharmaceutical spending, and it is contributing not only to containing spending but also to reducing it in the case of certain products (e.g., absorbents and test strips). In addition to these outcomes, it should also be noted that the development of nurse prescribing has been carried out to the full satisfaction of the professionals involved and the public [19].

### 1.1. General Objective

To determine the effectiveness of collaborative nurse prescribing in the follow-up and management of oral treatment with antivitamin K anticoagulants in PC compared with that of the traditional approach to the follow-up carried out by a family physician.

#### Specifics Objectives

-To evaluate the impact on the number of days of treatment that the patient is in therapeutic range.-To account for adverse events related to the procedure.-To describe adverse events related to the procedure.-To establish the economic impact on chapter I (personnel expenses).-To quantify the consumption of test strips for INR determination.

## 2. Materials and Methods

### 2.1. Research Hypothesis

Collaborative nurse prescribing (NP) in the follow-up and management of patients taking antivitamin K (AVK) oral anticoagulants in PC is more effective than the traditional approach to a follow-up carried out by a family physician.

### 2.2. Design

A multicenter, multidisciplinary (medicine and nursing) experimental analytical study will be conducted with two arms, one cohort (OAT) managed by family nurses and another cohort (OAT) managed by family physicians. This paper will be written in accordance with the CONSORT 2010 statement’s guideline for reporting randomized clinical trials and SPIRIT 2013.

### 2.3. Inclusion Criteria

Patients undergoing stabilized treatment with acenocoumarol or warfarin, for all authorized indications of use, referred to a PC professional for a protocolized follow-up. Ages between 30 and 95 years old. Acceptance of a follow-up by a physician or nurse being undertaken according to the advanced practice protocol for the follow-up of OAT patients of the Regional Ministry of Health and Consumer Affairs [12]. Based on the Spanish version of the Pfeiffer test, a cognitive capacity below 3 or 4 depending on whether or not they can read and write (Appendix A).

### 2.4. Exclusion Criteria 

Immobilized patients, with bed–chair life, who cannot go to the health center for the determination of INR, which is carried out at home by a nurse. Clinical changes, adverse effects and assumptions about the patient related to any of the contraindications stated in the protocol [12].

### 2.5. Study Population and Sample

The study population comprises 1208 anticoagulated patients registered in 2019 in TaoNet^®^ V 4.2 [20] (Roche Diagnostics software enabled by the Regional Ministry of Health and Consumer Affairs of the Andalusian Regional Government) for the Alamillo-San Jerónimo Clinical Management Unit. G*Power^®^ 3.1.9.4 software [21] is used to calculate the sample size, with an alpha error of 0.05, a power of 99% and an effect size of 0.5. The result is 127 participants per group. A total sample of 254 participants is needed. However, the universal sample from both health centers is used as follows: 606 patients are registered, 367 at Alamillo Health Center and 239 at San Jerónimo Health Center. The final sample is 575 participants (Figure 1). By means of a sequence executed in Excel^®^ (Redmond, Washington, DC, USA), the sample is randomized into a control group and an experimental group one month before the start of the experiment for those who agree to participate. The informed consent is completed and the Pffeifer test is performed. The study is blinded for participants as well as for researchers analyzing the data. Since OAT consultations are triaged and nurses who treat OAT patients in both centers participate in the study, stratification in clusters is not deemed necessary since randomization guarantees the homogeneous distribution of the entire anticoagulated population that meets the inclusion criteria, not just of the sample, from the two centers chosen in both groups.

### 2.6. Variables and Data Collection

The following study variables are used: patient age (years), biological sex of the patient (male/female), pathology for which VKA therapy is indicated, follow-up time (days from start of prescription), treating healthcare professional (nurse/physician), health center (Alamillo/San Jerónimo), anticoagulant drug (warfarin/acenocoumarol), INR (measured with Coagucheck^®^ coagulometer [22]), dosage (milligrams), test strips (number of strips per year × unit price: 3.87 EUR/stripe), time in therapeutic range based on pathology and prescriber indication (days), time out of therapeutic range (days), visits to health center (days), time spent per professional (minutes) and adverse events (hospitalization for out-of-range values related to OAT, clinical report of either hemorrhagic or thromboembolic process related to the last INR and medical/nursing consultation related to symptoms compatible with an hemorrhagic process). Variables are collected in an Excel^®^ data sheet by exporting the data from TaoNet^®^. The information regarding the presence of adverse events is obtained from the patient’s clinical history and the patient safety observatory records relating to the OAT office in the corresponding clinical management unit.

### 2.7. Procedure

The study is blind. An external agent assigns patients to each group of healthcare professionals, omitting such information to the research team. Nurses know which group patients belong to through whether or not they are responsible for their dosing. Patients are randomized and do not know whether they are assigned to the experimental or control group. The principal investigator and data analyzer are unaware of the patient assignment. The procedure is carried out between January and December 2020.

#### 2.7.1. Established Procedure of Follow-Up and Intervention

The patient visits the OAT office in Alamillo Health Center where they are seen by a physician and a nurse. The nurse determines the INR with an approved Coaguchek^®^ portable coagulometer. They communicate the result to the physician who enters the data into TaoNet^®^ and establishes the appropriate dose for the patient. The nurse delivers the resulting dosing schedule and schedules the patient for the next visit. If the patient is in range, an appointment is made in four to five weeks as in the previous two visits. If the range is altered, an earlier appointment is made proportionally to the INR alteration and at the discretion of the dosing clinician.

The patient visits the OAT office in San Jerónimo Health Center, as they do in Alamillo, except that in this case it is the nurse who enters the data into TaoNet^®^, which incorporates an automatic dosing algorithm. They print and deliver the dosing schedule without the intervention of a physician. If the INR is not in range, the software cannot calculate the dose automatically, and it is the physician who is responsible for dosing following the guidelines of the anticoagulation protocol of the referral hospital.

#### 2.7.2. Experimental Procedure of Follow-Up and Intervention

The patient visits an OAT office where they are seen by a nurse certified by the Andalusian Agency for Healthcare Quality (ACSA) for the follow-up of anticoagulated patients. The nurse determines the INR and inputs the data into TaoNet^®^. If the requirements for automatic dosing are met, they print the dosing schedule and schedule the patient for the next visit. If the requirements are not met, the nurse is responsible for varying the prescribed dose and rescheduling the patient based on the decision support algorithms outlined in the protocol that is the subject of this study [12]. This protocol states that the nurse is responsible for the dosing schedule. The nurse’s clinical experience in the management of these drugs is leveraged for decision making. This is shown in Figure 2 and Figure 3.

### 2.8. Data Analysis

The analysis to be performed include a descriptive analysis of qualitative variables (relative frequencies) and quantitative variables (measures of central tendency, dispersion and position); an exploratory analysis to identify the distribution of variables; a comparison between groups (bivariate analysis of qualitative variables with Pearson’s chi-square tests and quantitative variables through Student’s *t*-test for independent groups after checking normality) and comparison of two or more quantitative variables by means of ANOVA tests (normal distribution) or Shapiro–Wilk tests (non-normal distribution) with non-parametric tests such as the Kruskal–Wallis test. The significance level (*p*-value) determined will be <0.05 and the statistical software used will be the SPSS Statistics 25 package for Windows^®^ (Redmond, Washington, United States). The personnel expenses are calculated based on the updated data of chapter I provided by the Management Department of Sevilla District, the information on test strip consumption obtained from the SIGLO warehouse management software and the data provided by the Financial Department of Sevilla District.

### 2.9. Validity and Reliability

The results of the study can be influenced by several factors as follows: age, sex, follow-up time, level of understanding of instructions by patients, adherence to treatment or type and bioavailability of the drug. For this reason, a study design that minimizes confounding bias was used. The design randomly assigns patients to either group.

### 2.10. Ethical Considerations

This study applies the ethical principles of the Declaration of Helsinki and the principles of autonomy, non-maleficence, beneficence and justice of the Belmont Report. Informed consent is obtained from the participants (Appendix B) who are guaranteed the right to participate or not in the study, to leave the study at any time and to have the best available treatment proven (standard protocol intervention). The confidentiality and professional secrecy of the data obtained are preserved. Such information has been used only for the purposes of this study. The project has been approved by the relevant Ethics Committee (0331-N-19), (Appendix C).

### 2.11. Limitations

The effect of the intervention may be affected by adherence to treatment, which may be influenced by a series of aspects that are not measured in the study such as sociocultural level or income as well as socioeconomic variables. These may influence the decision to implement the recommendations of healthcare providers to a higher or lesser degree. The randomization of participants tries to prevent these conditioning factors from influencing the results of one group or the other.

## 3. Discussion

Studies with warfarin indicate good patient adherence (74%) and compliance with the therapeutic regimen (51%), which are important when considering making changes to the regimen. Patients with good adherence to warfarin have good compliance with other therapeutic regimens, such as diet, exercise and INR control, and show a willingness to cooperate with healthcare services [23]. 

OAT is currently undergoing a process of change due to the appearance in recent years of direct-acting anticoagulants (DOACs) that do not require analytical control [13,19]. This situation may make us think of a significant decrease in the number of anticoagulated patients who will stop taking VKAs in the future to mostly take DOACs.

Cost studies in Spain assessing apixaban versus acenocoumarol for stroke prevention in a cohort of 1000 patients with non-valvular atrial fibrillation (NVAF) reveal cost-effectiveness, as well as an increase in life expectancy and quality-adjusted life years. The economic model of Dorian et al. and a Markov model were adapted to the Spanish healthcare environment. When evaluating results, it must be taken into account that they are based on a theoretical model and thus represent a simplified simulation of reality by definition [24].

Two economic analyses of oral anticoagulants compared with acenocoumarol for stroke prevention in Spain [25,26] indicate that the avoided cost per stroke would be 17,584 EUR with dabigatran [27] and 11,274 EUR with rivaroxaban [26].

Two cost-effectiveness analyses of apixaban versus warfarin in non-valvular atrial fibrillation have been published in the United States (USA) according to which apixaban would be cost-effective in 98% and 62% of simulations for primary and secondary stroke prevention, respectively [28,29].

A study comparing acenocoumarol with dabigatran for stroke prevention in AF concluded that there was an improvement in the events suffered and gains in quantity and quality of life using a Markov model that simulates the natural history of 10,000 patients. As in the study by Baron et al., this represents a simulation. The high cost of dabigatran (1337 EUR per patient a year) compared with acenocoumarol (300 EUR per patient a year) means that in a more realistic scenario, the change can only be associated with higher-risk or difficult-to-control patients [30].

Our study takes into account the adverse events associated with VKAs. The switch to DOACs involves considering the risks they pose, including similar side effects related to bleeding in various locations. Renal elimination must be controlled at the beginning of and during treatment (annually) [31]. Poor renal function is a factor to consider in decreased efficacy [23].

In the USA, it is estimated that between 2 and 4 million people had serious or very serious adverse events related to drug treatments, most frequently with dabigatran and warfarin. In such an event caused by VKAs, the oral or intravenous administration of vitamin K can solve a potential bleeding problem [31]. In the first case, the approach is carried out in PC.

With DOACs, the intervention is more complex. Except for dabigatran, which has an inhibitor, the rest of the DOACs require a modified factor XA molecule, both of which are hospital-based solutions [32].

Given the limited experience of use in contrast with dicoumarimics, DOACs raise uncertainties about adverse events, especially about that of a possible increase in myocardial infarction as compared with dicoumarimics [33]. Renal function controls with DOACs [31] or decreased efficacy with decreased renal function are factors to consider [34].

There may be less adherence to DOACs considering that it is more likely to forget a dose when the number of doses is higher (two) and also that such patients are not as closely monitored as those with INR follow-ups [35].

In both centers that are the subject of this study, a pilot study was carried out through a prospective longitudinal (single-blind) experimental analytical study. Data collection was carried out for 4 months from January to April 2016 [36].

The calculation of the sample size, with a confidence level of 95%, precision level of 3% and an adjusted to losses level of 20%, was 200. For the study, we worked with a total of 147 patients as follows: 74 in the experimental group and 73 in control, using randomized patients from whom informed consent was obtained. The following results were obtained.

For the variable “days in range”, we made measurements through Student’s *t*-test and obtained the result that the average number of days that patients who were in range in the experimental group were greater than those in the control group. 

For the variable “days out of range” we obtained the result that the average number of days out of range for the experimental group was lower than the average for the control group. The U value for days out of range was 682.5 with a significance of (*p* = 0.0001).

With Spearman’s Rho, we verified that there was no significant statistical relationship (*p* = 0.0939) between the days from the start of anticoagulant treatment and the measured INR values.

The differences by sex found in both groups and also in age were analyzed to see if they were related to the variable “being in range” with *p* = 0.015 and the effect size of 0.19. The odds ratio did not indicate that male patients were more likely (2.4) to be in range than female patients [36].

## 4. Conclusions

The proposed study deepens the competency development of nurses and facilitates teamwork in PC. Nurses trained in the implementation of certain protocols can make clinical decisions previously only made by physicians. The literature consulted analyzing the substitution of physicians for nurses indicates that they can do it just as well, with no increase in adverse events. In Spain in general and in Andalusia in particular, collaborative prescribing protocols are being developed. There are no studies that have analyzed the outcomes with respect to the objectives set out in this project when nurses manage vitamin K antagonists in PC according to the current protocol.

## Figures and Tables

**Figure 1 healthcare-12-00347-f001:**
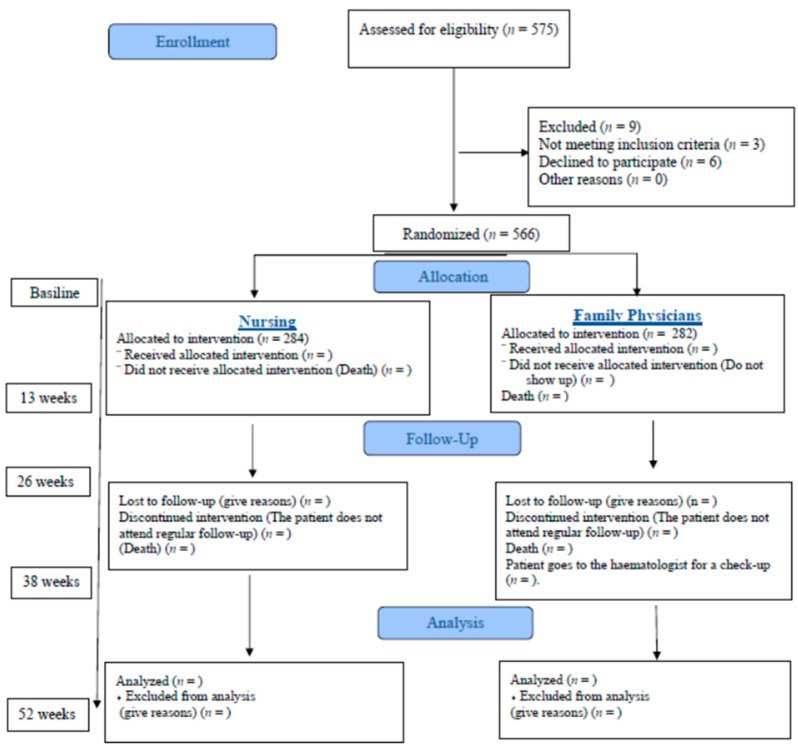
Flow diagram protocol.

**Figure 2 healthcare-12-00347-f002:**
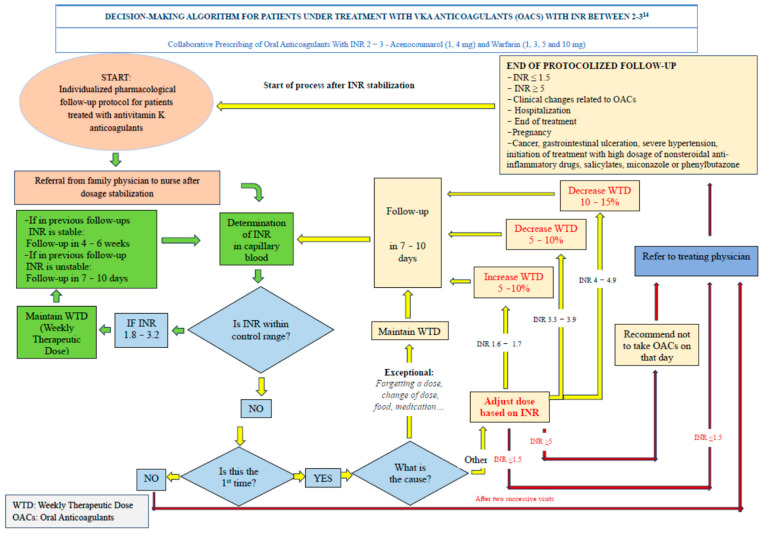
Decision-making algorithm for patients under treatment with vka anticoagulants with INR between 2–3.

**Figure 3 healthcare-12-00347-f003:**
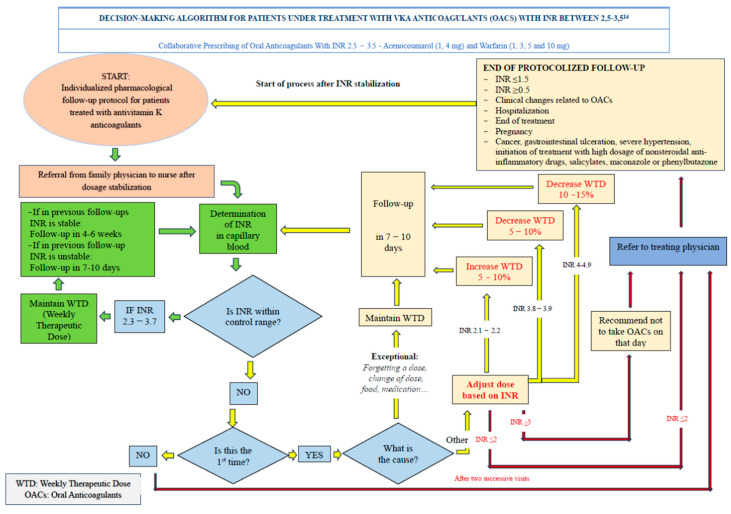
Decision-making algorithm for patients under treatment with vka anticoagulants with INR between 2.5–3.5.

## Data Availability

The data presented in this study are available on request from the corresponding author.

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
