# Peer review of "Effectiveness and Health Outcomes of Collaborative Nurse Prescribing for Patients Anticoagulated with Antivitamin K in Primary Care: A Study Protocol"

_healthcare, 2024, doi:10.3390/healthcare12030347_

Round 1

Reviewer 1 Report

Comments and Suggestions for Authors

REVISION: COMMENTS AND SUGGESTIONS

1. The autors cited to “Horrocks et al. whose conducted a systematic review in 2002 in which they concluded that increasing the availability of PC nurses was likely to improve patient satisfaction and the quality of care received”, Are there more studies from this systematic review or has this need been raised since 2002 and there have been no other studies? If there are any, it would be important to link them in the writing since it seems that there were no other studies that would expand on this one found in 2002.

2. The most recent reference is from 2017, are there no more studies in 6 years?

3. The previous question or question 3 is also for the authors to explain the current use of technologies and different digital applications, given that the studies cited in this study use the telephone medium given the years in which they were carried out. Since we are talking about efficient protocols, the way it is communicated is part of this efficiency.

4. References 28 and 29 are suggested to review and enter the year of consultation and 28 cannot be opened. References 33 and 34, what is the justification for placing them?

5. Could the authors explain why they recruited more study participants than established in the sample size calculation; Likewise, mention if the criterion for calculating the sample size was the studies or previous evidence mentioned in the introduction section.

6. What was the period of the study? And who applied informed consent and at what point in the study?

7. What was the statistical parameter considered to measure the effectiveness of the intervention? A number needed to treat was shown?

8. What were the control variables for this study, for example, age, education, years of having the disease, family support, socioeconomic level? Although it is mentioned in lines 246 to 256, point out why not run a control analysis of these variables if several of them were collected and why was it not demonstrated from the beginning? Did the findings or previous evidence not take them into account either?

9. Why is there no results section?

10. The discussion requires the review of the results, although tables are attached, why not develop this section?

11. The conclusion does not agree with the objective set by the authors, a type of recommendation appears.

12. The authors are asked to clarify whether these findings are intended to incorporate any modification of the normed or regulated procedures for the management of these patients in the hospitals where they were recruited.

13. The study needs the results of each objective and hypothesis proposed since the most important thing is not available. The application of these findings or their reproducibility is not clear due to the lack of methodological information.

Author Response

REV 1

  1. The autors cited to “Horrocks et al. whose conducted a systematic review in 2002 in which they concluded that increasing the availability of PC nurses was likely to improve patient satisfaction and the quality of care received”, Are there more studies from this systematic review or has this need been raised since 2002 and there have been no other studies? If there are any, it would be important to link them in the writing since it seems that there were no other studies that would expand on this one found in 2002

Yes, several articles related to the intervention are included.

  1. The most recent reference is from 2017, are there no more studies in 6 years?

Yes, new articles have been included as references.

  1. The previous question or question 3 is also for the authors to explain the current use of technologies and different digital applications, given that the studies cited in this study use the telephone medium given the years in which they were carried out. Since we are talking about efficient protocols, the way it is communicated is part of this efficiency.

In the protocol that is carried out the intervention is in a personal manner with a nurse

  1. References 28 and 29 are suggested to review and enter the year of consultation and 28 cannot be opened. References 33 and 34, what is the justification for placing them?

Reviewed: Reference 33, currently 39, refers to a research book recommending that both positive and negative research outcomes should be published. Reference 34, currently 40, refers to the Data Protection Law of the country where the research is conducted, whose guidelines should be considered.

  1. Could the authors explain why they recruited more study participants than established in the sample size calculation; Likewise, mention if the criterion for calculating the sample size was the studies or previous evidence mentioned in the introduction section

We have selected the entire population from both centers under Primary Care follow-up to participate in the study as we believe that the results are more representative with the largest possible sample.

In the new bibliography we have included three different studies: on the one hand there is a study where we have worked with 20 patients. On the other and we have another study where we have included 107 patients in the experimental group and 116 in the control one; considering that our study should have as many participants as possible.

  1. What was the period of the study? And who applied informed consent and at what point in the study?

From January to December 2020. The informed consent was administered by the principal investigator two months before each participant was included in the study.

  1. What was the statistical parameter considered to measure the effectiveness of the intervention? A number needed to treat was shown?

To assess the effectiveness of the intervention, the total number of visits for each patient during the study period is considered. Of these visits, the number within the therapeutic range and outside of it are considered. Since each visit incurs the cost of a test strip, the effectiveness of the intervention will be determined by the group whose patients have more visits within the therapeutic range, a lower number of visits in a year, and a lower expenditure on test strips in a year.

  1. What were the control variables for this study, for example, age, education, years of having the disease, family support, socioeconomic level? Although it is mentioned in lines 246 to 256, point out why not run a control analysis of these variables if several of them were collected and why was it not demonstrated from the beginning? Did the findings or previous evidence not take them into account either?

Those data were not considered. Since membership in the experimental or control group was completely random and blind to the patients, the possible influence of any of these factors on its outcome is randomly distributed in both groups.

  1. Why is there no results section?

There are no results as it is a protocol for an intervention before the execution of the described experiment. There are data from the previous pilot study.

  1. The discussion requires the review of the results, although tables are attached, why not develop this section?

As it is a protocol for an intervention before an experimental study, we are not able to know what results we will obtain.

  1. The conclusion does not agree with the objective set by the authors, a type of recommendation appears.

If the study is positive as it was in the pilot study, we can replace doctors by nurses in a good part of the follow-up of patients on VKA.

  1. The authors are asked to clarify whether these findings are intended to incorporate any modification of the normed or regulated procedures for the management of these patients in the hospitals where they were recruited.

Yes, since it is an official protocol, a positive result of the study provides evidence for replacing doctors by nurses in the follow-up of patients taking VKA.

  1. The study needs the results of each proposed objective and hypothesis since the most important thing is not available. The application of these findings or their reproducibility is not clear due to the lack of methodological information.

Since it is a protocol for an intervention before a study, there are no results or discussions possible. The results of the study will be the subject of another publication in which we can describe the obtained results.

Reviewer 2 Report

Comments and Suggestions for Authors

Dear authors,

It was a pleasure for me to review this manuscript that tries to compare the effectiveness and results of collaborative nursing prescription in patients anticoagulated with vitamin K in primary care.

I found the topic very interesting since there are more and more patients with chronic diseases and nurses are a key element in being able to provide these patients with quality health care. Unfortunately, in many cases, the primary care system is overwhelmed, which makes it more difficult for these chronic patients to access health services.

With the sole objective of improving the quality of the manuscript, I will allow myself to make a series of comments:

1. Line 20: The acronym NP is used without having previously described it.

2. Line 79: it is said that these protocols are already being carried out in other countries. I think it would be interesting to name some of the countries where these models are being applied.

3. Objectives. Normally in scientific articles, the objectives of the study are usually put at the end of the introduction section and not in the material and methods section. I suggest moving the objectives to the introductory section.

4. Likewise, I think that the description of the research hypothesis and the null hypothesis is not appropriate to put in the methodology section. This description is more typical of academic works than scientific articles.

5. Section 2.7 does not say the number of subjects that make up the study population. It is said in the summary but it does not appear in this section.

6. In section 2.7 it is said that the universal sample is made up of 606 registered patients but a final sample of 498 participants was considered. Please, I think it would be interesting to explain more clearly how the final sample was selected. It wouldn't hurt to consider putting up a flow chart that explains it.

7. In the methodology section, a point called ethical considerations must be added. These considerations should be put here and explain which ethics committee authorized the conduct of this study.

8. The sample was selected, the groups were made, but however the results are not presented. I agree that this is a study protocol, but from 2020 to 2023, no type of results are collected? When will the study conclude?

Thanks

Author Response

  1. Line 20: The acronym NP is used without having previously described it.

Revised

  1. Line 79: it is said that these protocols are already being carried out in other countries. I think it would be interesting to name some of the countries where these models are being applied.

Included

  1. Objectives. Normally in scientific articles, the objectives of the study are usually put at the end of the introduction section and not in the material and methods section. I suggest moving the objectives to the introductory section.

Rectified

  1. Likewise, I think that the description of the research hypothesis and the null hypothesis is not appropriate to put in the methodology section. This description is more typical of academic works than scientific articles.

Rectified

  1. Section 2.7 does not say the number of subjects that make up the study population. It is said in the summary but it does not appear in this section.

Rectified. Figure 1.

  1. In section 2.7 it is said that the universal sample is made up of 606 registered patients but a final sample of 498 participants was considered. Please, I think it would be interesting to explain more clearly how the final sample was selected. It wouldn't hurt to consider putting up a flow chart that explains it.

Figure 1.

  1. In the methodology section, a point called ethical considerations must be added. These considerations should be put here and explain which ethics committee authorized the conduct of this study.

Included point 2.10.

  1. The sample was selected, the groups were made, but however the results are not presented. I agree that this is a study protocol, but from 2020 to 2023, no type of results are collected? When will the study conclude?

Study concluded in 2022, we intend to publish the results

Reviewer 3 Report

Comments and Suggestions for Authors

Dear authors,

Please below:

Abstract: The structure is poor, need to rewrite the paragraph. The section starts with one aim and finishes with another aim. Abstract need to provide a a short summary of the article, starting with a brief introduction, aims, results and conclusion. 

Analysis: A plethora of tests have been used, which I believed its an excellent point since each software examines a different parameter. However, 2 tests have been used for comparison 'comparison of two or more quantitative variables by means of ANOVA tests (normal distribution) or Shapiro-Wilk tests' please explain the reason of having two. 

Results should be presented in a big table. The absence of table affects the general presentation of the paper. Currently , it has only text.

The appendix is too long, Can  the diagrams be removed to main section? 

Overall presentation, flow and structure needs editing and re-writing. 

Comments on the Quality of English Language

Please check and correct the paragraphs that contain short sentences. That will improve further the quality of paper.

Author Response

Abstract: The structure is poor, need to rewrite the paragraph. The section starts with one aim and finishes with another aim. Abstract need to provide a a short summary of the article, starting with a brief introduction, aims, results and conclusion. 

It is a protocol prior to the execution of the experiment se we are not able to indicate its results yet. Our aim is to demonstrate the study hypothesis.

Analysis: A plethora of tests have been used, which I believed its an excellent point since each software examines a different parameter. However, 2 tests have been used for comparison 'comparison of two or more quantitative variables by means of ANOVA tests (normal distribution) or Shapiro-Wilk tests' please explain the reason of having two. 

We are unaware of the normality of the sample until the intervention is carried out, which is why we specify the statistical test based on the results obtained.

Results should be presented in a big table. The absence of table affects the general presentation of the paper. Currently, it has only text.

It is a study protocol preceding an experiment, so it is the reason why we do not have results yet. The statistical data in the discussion are derived from the literature, from a previously published pilot study that we conducted.

Palomo-Lara, J.C. Optimizing clinical management in a health care unit: an innovative experience. Rev ROL Enferm 2017; 40(9): 578-584

The appendix is too long, Can the diagrams be removed to main section? 

It has been already removed.

Round 2

Reviewer 2 Report

Comments and Suggestions for Authors

Dear authors,

It was a pleasure for me to review this second improved version of the manuscript.

I would like to congratulate the authors for the great effort made to improve this second version taking into account all the recommendations of the different reviewers. I consider that the result has been very good.

However, I have detected that there are errors in the bibliography. There are 38 citations in the text and 40 in the reference list. Furthermore, the references do not faithfully follow the Vancouver style. For example there is “;“ after the authors' initials. In many others, the Vancouver style is not followed to express volume and issue (reference 2, for example)

I congratulate the authors for the changes they have had to make to improve the version of the protocol and I encourage them to carefully review the bibliographic style.

Thank you so much.

Kind regards.

Author Response

reviewed and corrected the bibliography, thank you very much

Reviewer 3 Report

Comments and Suggestions for Authors

Dear Authors,

All issues addressed, thus recommending acceptance.

Author Response

Thank you very much for your comments; they have improved the wording of the article